# Discovery of Dolutegravir Derivative against Liver Cancer via Inducing Autophagy and DNA Damage

**DOI:** 10.3390/molecules29081779

**Published:** 2024-04-13

**Authors:** Xixi Hou, Dong Yan, Ziyuan Wu, Longfei Mao, Huili Wang, Yajie Guo, Jianxue Yang

**Affiliations:** 1The First Affiliated Hospital, and College of Clinical Medicine of Henan University of Science and Technology, Luoyang 471003, China; lucyfly881104@163.com; 2College of Basic Medicine and Forensic Medicine, Henan University of Science and Technology, 263 Kaiyuan Road, Luoyang 471003, Chinawuziyuandyx@163.com (Z.W.); longfeimao1988@163.com (L.M.); 3University of North Carolina Hospitals, 101 Manning Dr, Chapel Hill, Orange County, NC 27599, USA; huili.becker@gmail.com; 4Department of Emergency, The Eighth Affiliated Hospital, Sun Yat-sen University, Shenzhen 518033, China

**Keywords:** dolutegravir, 1,2,3-triazole, anti-tumor, hepatocellular carcinoma cell lines DNA damage

## Abstract

We introduced a terminal alkyne into the core structure of dolutegravir, resulting in the synthesis of 34 novel dolutegravir-1,2,3-triazole compounds through click chemistry. These compounds exhibited remarkable inhibitory activities against two hepatocellular carcinoma cell lines, Huh7 and HepG2. Notably, compounds **5e** and **5p** demonstrated exceptional efficacy, particularly against Huh7 cells, with IC_50_ values of 2.64 and 5.42 μM. Additionally, both compounds induced apoptosis in Huh7 cells, suppressed tumor cell clone formation, and elevated reactive oxygen species (ROS) levels, further promoting tumor cell apoptosis. Furthermore, compounds **5e** and **5p** activated the LC3 signaling pathway, inducing autophagy, and triggered the γ-H2AX signaling pathway, resulting in DNA damage in tumor cells. Compound **5e** exhibited low toxicity, highlighting its potential as a promising anti-tumor drug.

## 1. Introduction

Human immunodeficiency virus (HIV) is a lentivirus that infects cells of the human immune system. HIV is a member of a class of retroviruses which was identified in the US in 1981. By damaging the body’s immune system, the virus causes a variety of diseases including cancer, and ultimately threatens the lives of patients [1]. After a patient is infected with HIV, the virus cannot be completely eliminated from the body and we can only rely on drug treatment to reduce the viral load in body [2]. HIV integrase inhibitors are an important class of anti-AIDS drug, which can inhibit the replication process of the retrovirus and block the integration of virus DNA and host chromosome DNA [3,4]. Combination therapy with other anti-retroviral drugs, such as Entecavir, effectively treats HIV infection, reduces drug resistance, significantly improves the quality of life and survival time of patients with AIDS, and reduces mortality [5,6]. However, AIDS patients are prone to liver damage when receiving drug treatment, especially when multiple drugs are used in combination. Research has found that liver disease has gradually become the most common complication in AIDS mortality, accounting for 14% to 18% of all AIDS deaths [7,8,9]. Liver damage in AIDS patients mainly includes abnormal liver function, liver failure, liver fibrosis, liver cirrhosis, primary liver cancer, and end-stage liver disease [10,11]. Therefore, the structural modification of anti-HIV drugs for the function of treating liver diseases, and especially the ability to inhibit the activity of liver cancer cells, has research significance.

Dolutegravir (DTG, Figure 1), as a class of HIV integrase inhibitors approved by FDA priority [12], has strong anti-viral and anti-drug-resistant properties [13]. In the treatment of patients with first-time HIV infection, DTG taken once a day is comparable to Raltegavir (RAL) taken twice a day. The results of preclinical study show that DTG has little toxicity. When the dose of DTG was 27 times greater than the clinical dose, no obvious fertility toxicity or teratogenic toxicity was found. The results of clinical studies show that DTG is better than a control drug in the treatment of HIV first-infected people and has better effect on patients who failed treatment without use of integrase inhibitors. Good responses were also found in adult patients who were resistant to raltegravir (RAL) or elvitegravir (EVG) [14,15].

Lead discovery is one of the most important issues in drug research and discovery. Among several methods developed, using an already marketed drug as the starting point has been proved to be one of the most efficient methods in lead discovery. Already marketed drugs generally pass the toxicity test, show good solubility, and promising pharmacokinetic properties. Further, structure modification of these already marketed drugs may result in the formation of compounds which may be active against different targets, which means that they may be used in the treatment of different diseases. In view of its strong anti-HIV activity, good safety and tolerance, Dolutegravir was selected as the research object in this work. As an important class of nitrogen-containing heterocyclic compounds, 1,2,3-triazoles can be easily, efficiently and quickly prepared by click reaction [16]. 1,2,3-triazole derivatives are widely used in the modification of drug molecules because of their chemical properties, such as amide electron arrangement and stable rigid plane [17,18,19,20,21]. Therefore, a series of 1,2,3-triazoles derivatives was designed and synthesized by click reaction using dolutegravir as the parent nucleus according to the principle of bioactive sub-structure splicing. We used the CCK-8 method to evaluate the anti-proliferative activity of the target compounds on two kinds of hepatoma cell lines, Huh7 and hepG2, as well as the proliferative inhibitory activity of normal cell HRM.

## 2. Chemistry

A multi-step synthesis route was employed using 1-(2,2-dimethoxyethyl)-5-methoxy-6-(methoxycarbonyl)-4-oxo-1,4-dihydropyridine-3-carboxylic acid (**1**) as the starting material. The process involved the hydrolysis of compound **1** under formic acid to generate compound **2**. Subsequently, (*R*)-3-aminobutanol was introduced directly to the vacuum concentration, and the resulting mixture was refluxed in acetonitrile to yield compound **3** [22,23]. Compound **3** underwent condensation with 3-amine phenylacetylene, utilizing HATU and DIPEA, leading to the formation of the terminal alkyne compound **4**. The subsequent reaction of compound **4** with azide compounds, each bearing distinct substituents, resulted in the synthesis of 34 novel target compounds (**5a**–**5z** and **6a**–**6h**), as illustrated in Figure 2 and detailed in Table 1. The structures of the target compounds were confirmed through ^1^H and ^13^C nuclear magnetic resonance (NMR) spectroscopy. This versatile synthetic approach yields a diverse set of compounds, offering potential for further exploration of their biological activities.

## 3. Results and Discussion

### 3.1. Dolutegravir Derivatives Suppressed Cancer Cell Viability

This study aimed to assess the anti-proliferative activity of dolutegravir-1,2,3-triazole derivatives on hepatocellular carcinoma cell lines, Huh7 and hepG2. A comprehensive CCK8 assay was employed to evaluate the impact of these compounds on cell viability, with cells treated at a concentration of 20 μM for 48 h. HRM cells, a normal cell line, were included as a control for comparative analysis. The results, presented in Table 2, revealed significant anti-proliferative effects for most compounds on both Huh7 and hepG2 cell lines.

Further investigation focused on selected compounds that exhibited efficacy across multiple cell lines. The half maximal inhibitory concentration (IC_50_) was determined for compounds **5e** and **5p**, emerging as the most potent against Huh7 and hepG2 cells. The IC_50_ values for **5e** were 2.64 ± 0.47 μM and 5.42 ± 0.43 μM in Huh7 and hepG2 cells, respectively. Compound **5p** demonstrated IC_50_ values of 6.84 ± 0.99 μM for Huh7 and 4.83 ± 1.17 μM for hepG2 cells (Table 3). In fact, the IC_50_ values of these compounds against Huh7 cell and hepG2 cell are slightly different, and most are not very different. The reason for this difference may lie in the origin and characteristics of the two cell lines. The Huh7 cell line was established by Nakabayashi et al., derived from a Japanese male highly differentiated hepatocellular carcinoma [24]. Huh7 cells can produce some cytoplasmic proteins, such as albumin, antitrypsin, and AFP. It is characterized by HBV negative, and has hepatitis C virus susceptibility, which can be used to study carcinogenicity, gene expression regulatory mechanism, metabolism, and VLDL secretion. The HepG2 cell line is derived from liver cancer tissue of a 15-year-old white man from the Caucasus region [25]. The liver cancer tissue type is hepatoblastoma. hepG2 cells can secrete ALB and a2-MG and have a high degree of differentiation. The biotransformation characteristics of metabolic enzymes in the hepG2 cells are relatively complete. It can be used as an ideal cell line for in vitro hepatocyte metabolism and toxicity study. These findings underscore the potential of specific dolutegravir-1,2,3-triazole derivatives as promising candidates for further anti-cancer drug development.

### 3.2. Compounds ***5e*** and ***5p*** Inhibited Proliferation of Cancer Cells

To further assess the anti-proliferative activity of dolutegravir derivatives, LIVE/DEAD staining was conducted. Huh7 cells were treated with 5 μM or 10 μM of **5e** and **5p** for 24 h, followed by live and dead cell imaging and quantification. The results demonstrated a significant dose-dependent decrease in live Huh7 cells after treatment with **5e**. The ratio of dead to live cells also substantially increased with concentration (Figure 3A). Similarly, for **5p**, the number of live Huh7 cells decreased with increasing concentration. Although the ratio of dead to live cells increased compared to the untreated group, it did not show complete dose dependence. This may be attributed to significant cell proliferation suppression at 10 μM concentration, leading to a limited number of viable cells. Furthermore, to validate the impact of dolutegravir derivatives on cell proliferation, a plate clone formation assay was performed. Cells were exposed to various concentrations (0, 2, 4, 8, 16, and 32 μM) of **5e** or **5p**. Consistent with the above findings, both **5e** and **5p** exhibited dose-dependent anti-proliferative activity across Huh-7 cell lines (Figure 3B).

### 3.3. Compounds ***5e*** and ***5p*** Induced Apoptosis of Cancer Cells

Given the observed inhibitory effects of compounds **5e** and **5p** on cancer cell proliferation, we further investigated their impact on cell apoptosis. Apoptosis analysis was conducted on Huh7 cells treated with varying concentrations of **5e** or **5p**, utilizing Annexin V-FITC and PI staining, followed by flow cytometry to quantify apoptotic cells. Results revealed a notable increase in apoptosis in Huh7 cells treated with 8 μM of **5e** for 48 h, while no significant changes were observed at 2 μM or 4 μM of **5e** (Figure 4). Additionally, 4 μM and 8 μM concentrations of **5p** induced remarkable apoptosis in Huh7 cells, whereas 2 μM showed no significant impact (Figure 4).

### 3.4. Compounds ***5e*** and ***5p*** Changed Cell Cycle in Cancer Cells

To further investigate the effects of compounds **5e** and **5p** on regulation of the cell cycle, we treated cancer cells with different concentrations of **5e** or **5p** for 48 h and then analyzed cell cycle distribution using flow cytometry. As showed, 4 μM and 8 μM of **5e** could increase the S phase in Huh7 cells while 8 μM of **5p** showed a clearly decreased phase of S (Figure 5).

### 3.5. Compounds ***5e*** and ***5p*** Triggered Reactive Oxygen Species Generation in Cancer Cell Lines

Reactive oxygen species (ROS) play a crucial role in inducing cell death and inhibiting cell growth and proliferation. To investigate the impact of compounds **5e** and **5p** on ROS generation, various cancer cells were treated with **5e** or **5p** for 24 h. Subsequently, cells were stained with DCFDA [26], and ROS levels were visualized using a fluorescent microscope (Thermo, Beijing, China). As depicted in Figure 4, there was a significant increase in ROS generation observed after treatment with **5e** or **5p** in Huh7 cells (Figure 6).

### 3.6. Compounds ***5e*** and ***5p*** Affected Protein Expressions of Key Signaling Pathways

To elucidate the impact of compounds **5e** and **5p** on the regulation of cell proliferation, we examined the expression levels of key proteins involved in cell growth processes, encompassing autophagy, apoptosis, cell cycle, and DNA damage (Figure 7). Ubiquitin-like molecule light chain 3 (LC3), a pivotal marker of autophagy, exhibited a significant increase in expression following treatment with **5e** or **5p** in Huh7 cells. Moreover, treatment with **5e** and **5p** led to the induction of γ-H2AX, indicating that these compounds could induce DNA damage in Huh7 cells. However, the expression of Caspase3, a key protein in regulating apoptosis, remained unchanged in cancer cells treated with **5e** or **5p**. Similarly, cell cycle-related genes, including Cyclin D, Cyclin E, or β-catenin, showed no significant differences with treatment of **5e** or **5p** in Huh7 cells.

### 3.7. Compounds ***5e*** and ***5p*** Cause DNA Damage in Huh7 and HepG2 Cells

The phosphorylation of H2AX serves as a highly specific and sensitive molecular marker for DNA damage. Through Western blotting, we have confirmed that compounds **5e** and **5p** induce DNA damage in Huh7 and HepG2 cells. To further substantiate the inhibitory effect of compounds **5e** and **5p** on Huh7 cells, immunofluorescence was employed to assess their correlation with γ-H2AX. The results demonstrated that both **5e** and **5p** increased the expression of γ-H2AX in Huh7 and HepG2 cells, suggesting a relationship between the inhibitory effect, apoptosis induction, and DNA damage caused by the compounds (Figure 8).

### 3.8. In Vivo Toxicity of Compound ***5e*** to Mice

Through cytotoxicity experiments, we found that compound **5e** exhibited low toxicity, as it showed no inhibitory effect on human renal mesangial cells at a concentration of 20 μM. To further investigate the in vivo toxicity of **5e**, acute toxicity experiments were conducted. Twelve mice aged 6–8 weeks with body weights of 18–22 g were selected and divided into two groups, with three male and three female mice in each group. Compound **5e** was orally administered at a single dose of 500 mg/kg after being formulated in a carboxymethylcellulose solution. The mice were fasted for 12 h before dosing and allowed to consume a normal diet for 2 h after dosing. Observations and recordings were made for 12 days. On the final day, the mice were euthanized. As shown in Figure 9A, there were no deaths among the mice during the 12-day observation period, and the mice in both the treated group and the control group showed increases in body weight that were not significantly different. This indicates that Compound **5e** has low toxicity. Additionally, major organs were subjected to H and E staining. As shown in Figure 9B, after drug treatment, there was no apparent toxicity observed in the organs of the treated mice, such as the lung, liver, stomach and kidney, compared to those in the control group. Further acute toxicity experiments in mice have demonstrated the safety of compound **5e** for normal cells and tissues, indicating its potential as a lead compound for anti-tumor activity.

## 4. Experimental

### 4.1. Materials and Chemistry

Dolutegravir-1,2,3-triazole derivatives were synthesized in-house. Dimethyl-sulfoxide (DMSO) was obtained from Sigma-Aldrich (St. Louis, MO, USA). Dulbecco’s modified Eagle medium (DMEM), RPMI 1640 Medium, Fetal bovine serum (FBS) and penicillin/streptomycin were purchased from Gibco (Grand Island, NY, USA). Enhanced Cell Counting Kit-8, Calcein/PI Live/Dead Viability Assay Kit, Giemsa dye and Reactive Oxygen Species (ROS) Assay Kit were obtained from Beyotime Biotechnology (Shanghai, China). Annexin V-FITC/Propidium iodide (PI) staining kit and Matrigel Matrix were provided by BD Biosciences (Franklin Lake, NJ, USA).

#### General Synthetic Route of the Dolutegravir Derivatives

The general synthetic procedure from starting material 1-(2,2-dimethoxyethyl)-5-methoxy-6-(methoxycarbonyl)-4-oxo-1,4-dihydropyridine-3-carboxylic acid (compound **1**) to compounds **5a**–**5z** and **6a**–**6h** is the same as reported in our previous work [27]; details can be found in Appendix A.

### 4.2. Biological Study

#### 4.2.1. Cell Culture

Human hepatocellular carcinoma cell lines Huh7 and hepG2 were obtained from ATCC (Manassas, VA, USA). Cells were cultured in DMEM or RPMI 1640 medium containing 10% FBS and 1% penicillin/streptomycin at 37 °C with a 5% CO_2_-humidified atmosphere.

#### 4.2.2. Cell Viability Assay

CCK8 assay was used to measure cell viability. Cells with a density of 1 × 10^4^ cells/well were seeded on 96-well plates. After adhesion, cells were treated with different diluted compounds or vehicle control DMSO and continued to be cultured for 48 h or 72 h, respectively. Then, CCK8 reagent was added for one hour incubation at 37 °C with 5% CO_2_. Absorbance was measured using a Microplate spectrophotometer (Thermo, Waltham, MA, USA) at 450 nm. The ratio of cell viability of control was taken as 100%. For IC_50_, cells were treated with different concentrations of compounds (0, 0.5 μM, 2 μM, 8 μM, 16 μM, 32 μM) for 48 h and cell viability was determined to calculate the inhibition percentage. Then, IC_50_ of compounds were investigated using the prism statistical software 7.0.

#### 4.2.3. Live and Dead Cell Measurement

Huh7 cells with a density of 5 × 10^3^ cells/well were seeded on 96-well plates. Then, different concentrations (0, 5 μM, 10 μM) of **5e** or **5p** were treated for 24 h. Cells were then stained with the LIVE/DEAD Assay Kit, observed and photographed using the fluorescent microscope.

#### 4.2.4. Plate Clone Formation Assay

Huh7 cells were seeded into 6-well plates at a density of 100–500 cells/well. After 10 days culture, cells were added with **5e** or **5p** at different concentrations (0, 2 μM, 4 μM, 8 μM, 16 μM, 32 μM) for 48 h. Then, cells were fixed by 4% paraformaldehyde and stained by Giemsa dye. An optical microscope was used to photograph cells and the clone numbers were counted.

#### 4.2.5. Apoptosis Assay

Huh7 cells were cultured in 6-well plates with a density of 3 × 10^5^ cells/well. Different concentrations of **5e** or **5p** were added to cells for 48 h, respectively. The concentrations were 0, 2, 4, and 8 μM for Huh7 cells. After treatment, Annexin V-FITC Apoptosis Detection Kit (Elabscience, Wuhan, China) was used to determine the apoptotic ratio and FlowJo software v10 was used to analyze.

#### 4.2.6. Cell Cycle Assay

Huh7 cells were cultured in 6-well plates with a density of 3 × 10^5^ cells/well. Cells were treated with different concentrations (0, 4, and 8 μM for Huh7) of **5e** or **5p** for 48 h respectively. Then the cell cycle was determined by PI staining using the flow cytometer(Thermo, Beijing, China).

#### 4.2.7. Cellular ROS Measurement

Huh7 cells were cultured in 96-well plates in a density of 5 × 10^3^ cells/well. Different concentrations (0, 5, 10 and 20 μM) of **5e** or **5p** were added to cells, respectively, for 24 h. After treatment, cells were stimulated with 10 μM DCFH-DA for 30 min in 37 °C and then observed and photographed using the fluorescent microscope.

#### 4.2.8. Western Blot

Protein expression levels were measured by western blot. Huh7 cells were cultured in 12-well plates and different concentrations (0, 2, 4, 8 μM) of **5e** and **5p** were added for 48 h. Proteins were extracted from whole cells using radio-immunoprecipitation assay (RIPA) buffer containing a protease/phosphatase inhibitor cocktail (CST). Then, 10–15% sodium dodecyl sulfate polyacrylamide gel electrophoresis and nitrocellulose membranes (Millipore) were employed to separate and collect proteins. Antibodies used include LC3 (3868s, CST), Caspase3 (9662, CST), cyclin D (2922s, CST), cyclin E (20808s, CST), γH2AX (9718s, CST), β-Catenin (9562s, CST), and β-actin (4967s, CST).

#### 4.2.9. DNA Damage Staining

The Huh7 cells with a density of 5 × 10^3^ cells per well were seeded in a 96-well plate for overnight incubation. Then, different concentrations (0, 5 μM, 10 μM, 20 μM) of the compounds **5e** and **5p** were added to cells for 24 h. A DNA damage detection kit was used to dye the cells, and cells were observed and photographed by a fluorescent microscope. Cell nucleus is stained blue, and DNA damage is shown as green fluorescence. 

The HepG2 cells (1 × 10^6^) were plated in laser confocal dishes (Nest, Wuxi, China), before Compound **5e** and **5p** were treated for 72 h. Then, 4% paraformaldehyde was added to each laser confocal dishes for 20 min, after rinsing with PBS. The Triton X-100 was used to penetrate the cells for 20 min. Before the cells were incubated with p-H2AX (Cell Signaling Technology, Danvers, MA, USA) at 1:800 dilution at 4 °C overnight, goat serum was used to block for 30 min. Subsequently, the HepG2 were rinsed with PBS and incubated with Alexa Fluor 488 (Proteintech, Wuhan, China) at a 1:1000 dilution in the dark at room temperature for 1 h. DAPI was added to each laser to stain the cell nucleus for 5 min. Then, the cells were observed under a laser confocal microscope (Nikon, Tokyo, Japan), after rinsing with PBS.

#### 4.2.10. Statistical Analyses

Data were conducted using Graph Prim 7.0. A two-tailed Student’s *t*-test or one-way analysis of variance followed by a Student–Newman–Keuls (SNK) test were used to assess significant differences. Values of *p* < 0.05 were considered statistically significant.

## 5. Conclusions

In summary, our study highlights the successful synthesis of derivatives with distinctive 1,2,3-triazole moieties through conventional click reactions between dolutegravir and various azides. Notably, compounds **5e** and **5p** demonstrated significant anti-tumor activities, with IC_50_ values of 2.64 μM and 5.42 μM, respectively, against Huh7 cells. Subsequent investigations unveiled their capacity to induce apoptosis, hinder tumor cell clone formation, and elevate reactive oxygen species (ROS) levels, contributing to tumor cell apoptosis. Furthermore, both compounds activated the LC3 signaling pathway, inducing autophagy, and the γ-H2AX signaling pathway, inducing DNA damage in Huh7 tumor cells. Importantly, **5e** exhibited low toxicity against normal cells. Their ready availability, promising bioactivity against tumor cells, and relatively low toxicity against normal cells position these compounds as promising candidates for further structural optimization.

## Figures and Tables

**Figure 1 molecules-29-01779-f001:**
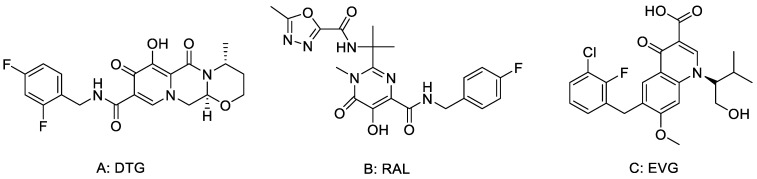
The structures of (**A**) DTG, (**B**) RAL and (**C**) EVG.

**Figure 2 molecules-29-01779-f002:**
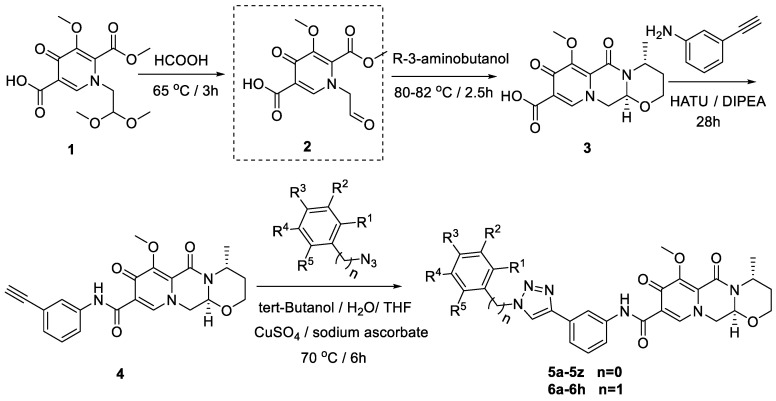
Reaction routes to compounds **5a**–**5z**, **6a**–**6h**.

**Figure 3 molecules-29-01779-f003:**
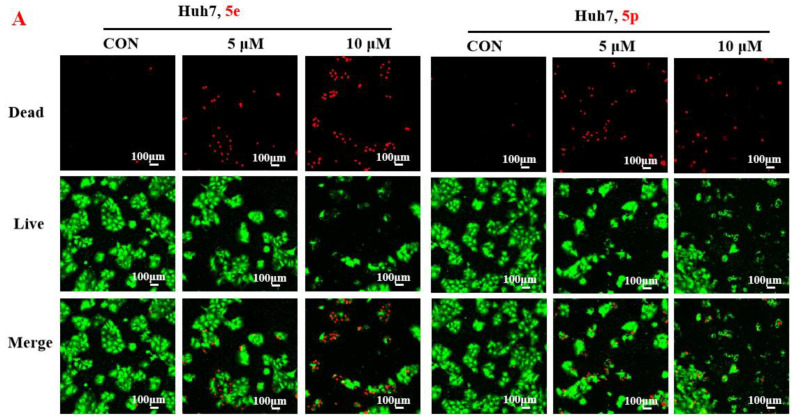
Compounds **5e** and **5p** inhibited proliferation of cancer cells. (**A**) Fluorescence images stained with the LIVE/DEAD kit of Huh7 cells treated with 5 μM and 10 μM of **5e** and **5p**. (**B**) Plate clone staining of Huh7 cells treated with different concentrations of **5e** and **5p**. Data are presented as mean ± SE. * *p* < 0.05.

**Figure 4 molecules-29-01779-f004:**
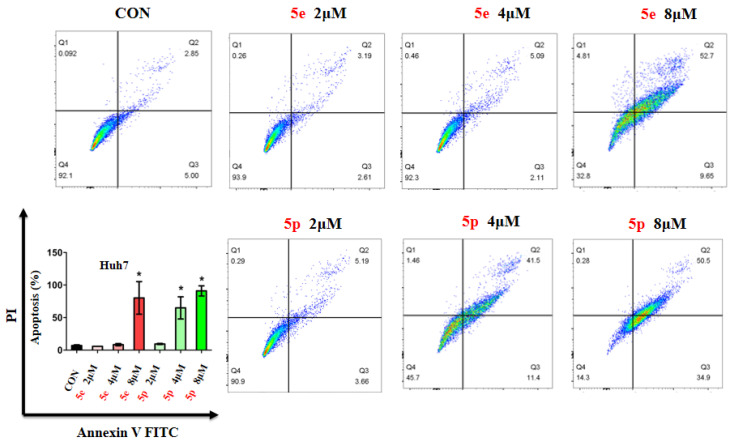
Compounds **5e** and **5p** induced apoptosis of cancer cells. Apoptotic cells of Huh7 cells treated with **5e** and **5p** determined by flow cytometry. Data are presented as mean ± SE. * *p* < 0.05.

**Figure 5 molecules-29-01779-f005:**
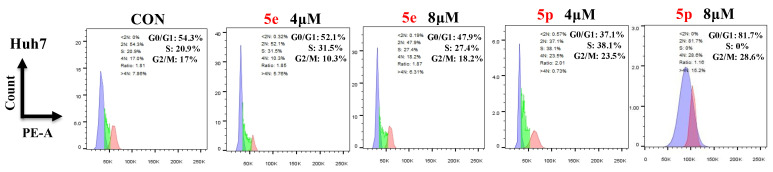
Compounds **5e** and **5p** changed cell cycle in cancer cells. Flow cytometry analysis of Huh7 cells treated with **5e** and **5p** compound for 48 h. Data are presented as mean ± SE.

**Figure 6 molecules-29-01779-f006:**
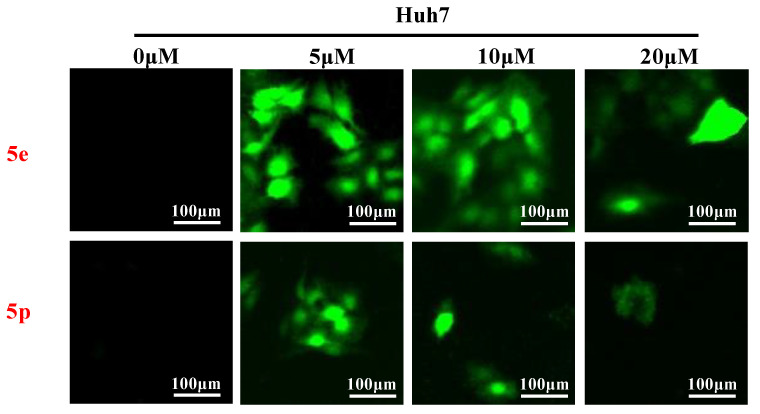
Compounds **5e** and **5p** triggered reactive oxygen species generation in cancer cell lines. ROS detection staining by DCFH-DA in Huh7 cells treated with **5e** and **5p** at the concentrations of 5 μM, 10 μM and 20 μM.

**Figure 7 molecules-29-01779-f007:**
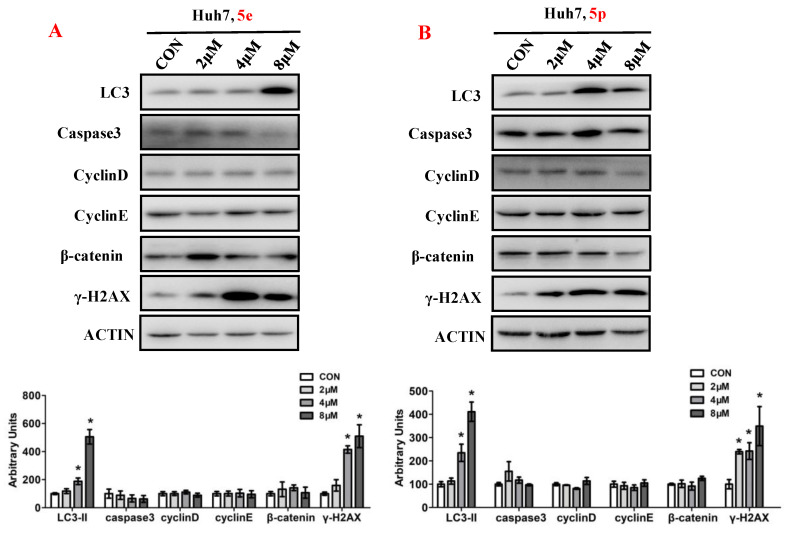
Compounds **5e** and **5p** affected protein expressions of key signaling pathways. (**A**) Western blotting of LC3, caspase3, Cyclin D, Cyclin E, γ-H2AX in Huh7 cells treated with **5e**. (**B**) Western blotting of caspase3, Cyclin D, Cyclin E, γ-H2AX in Huh7 cells treated with **5p**. Top, western blot; bottom, quantitative measurements relative to ACTIN. Data are presented as mean ± SE. * *p* < 0.05. The full-length original blot is included in the Appendix A. The samples were derived from the same experiment and the blots were processed in parallel.

**Figure 8 molecules-29-01779-f008:**
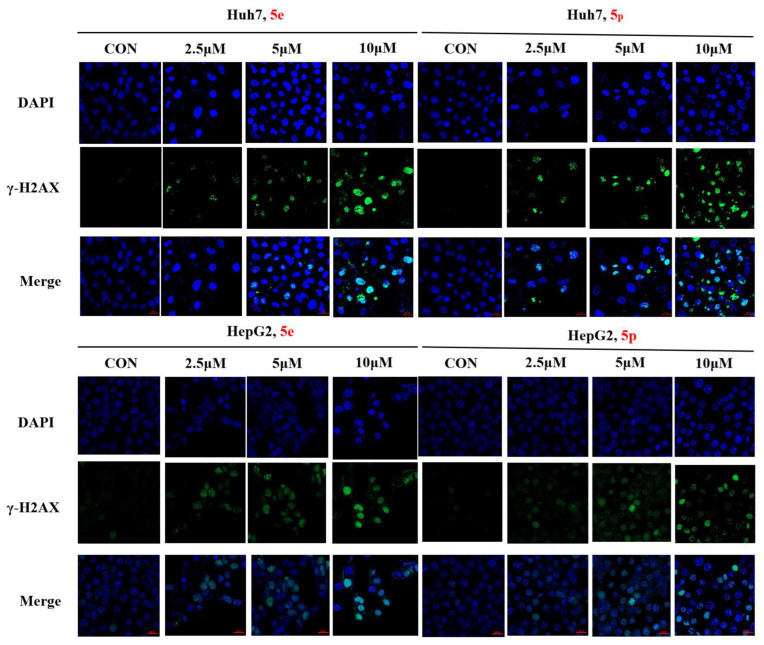
γ-H2AX phosphorylation induced by **5e** and **5p** in Huh7 and HepG2 cells by immunofluorescence. The scale bar is 20 μm.

**Figure 9 molecules-29-01779-f009:**
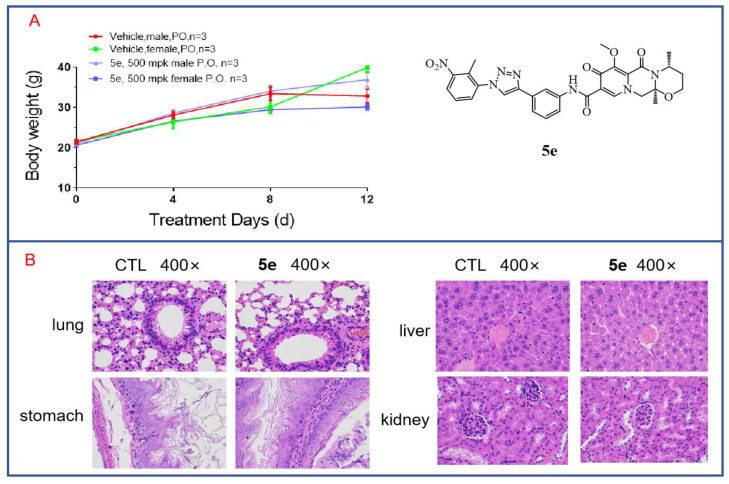
In vivo toxicity of compound **5e** in mice. (**A**) Acute toxicity experiments with compound **5e** were conducted in mice. (**B**) H and E staining was performed on various organs from mice treated with compound **5e**.

**Table 1 molecules-29-01779-t001:** R-group of compounds **5a**–**5z**, **6a**–**6h**.

Compd No.	R^1^	R^2^	R^3^	R^4^	R^5^
**5a**	F	H	H	H	H
**5b**	OCH_3_	H	NO_2_	H	H
**5c**	CF_3_	H	H	H	H
**5d**	H	H	CF_3_	H	H
**5e**	CH_3_	NO_2_	H	H	H
**5f**	H	CH_3_	H	H	H
**5g**	H	H	F	H	H
**5h**	H	H	OCH_2_CH_3_	H	H
**5i**	OCF_3_	H	H	H	H
**5j**	H	OCH_3_	OCH_3_	H	H
**5k**	CH_2_CH_3_	H	H	H	H
**5l**	H	OCH_3_	H	H	H
**5m**	H	H	CH_2_CH_3_	H	H
**5n**	H	H	C(CH_3_)_3_	H	H
**5o**	H	H	H	H	H
**5p**	H	CF_3_	H	H	H
**5q**	CF_3_	H	H	CF_3_	H
**5r**	H	CF_3_	H	CF_3_	H
**5s**	CH_3_	H	H	H	H
**5t**	OCH_3_	H	H	H	H
**5u**	Br	H	H	H	H
**5v**	Cl	H	H	H	H
**5w**	I	H	H	H	H
**5s**	H	H	OCH_3_	H	H
**5y**	H	H	CH_3_	H	H
**5z**	CH_3_	H	CH_3_	H	CH_3_
**6a**	H	Br	H	Br	H
**6b**	H	H	H	H	H
**6c**	Br	H	H	H	H
**6d**	H	OCH_3_	H	H	H
**6e**	H	Br	H	H	H
**6f**	F	H	H	H	H
**6g**	CH_3_	H	H	H	H
**6h**	CF_3_	H	H	H	H

**Table 2 molecules-29-01779-t002:** Inhibition performance of selected tumor cells by the compounds **5a**–**5z** and **6a**–**6h**.

Compd No.	Cell Viability (100%), 20 μM, 48 h		
	Huh7	hepG2	HRM
**5a**	83.32	90.43	71.18
**5b**	110.94	93.94	112.94
**5c**	36.28	54.14	2.65
**5d**	77.89	79.96	108.43
**5e**	33.91	39.65	85.04
**5f**	15.92	12.96	66.43
**5g**	90.24	105.19	103.67
**5h**	72.06	90.41	102.23
**5i**	7.95	24.84	70.03
**5j**	65.52	77.13	108.54
**5k**	48.55	50.62	0.81
**5l**	46.30	33.91	94.97
**5m**	36.41	77.30	101.13
**5n**	54.04	77.30	108.06
**5o**	67.64	80.21	100.28
**5p**	5.93	30.66	16.33
**5q**	12.00	38.49	91.40
**5r**	80.13	78.52	101.27
**5s**	61.76	69.61	100.81
**5t**	29.33	41.93	73.24
**5u**	33.46	47.16	100.10
**5v**	37.66	38.75	95.10
**5w**	49.87	54.35	92.77
**5s**	68.32	66.93	101.55
**5y**	79.52	69.40	103.82
**5z**	66.22	63.22	100.35
**6a**	75.97	66.48	84.46
**6b**	68.00	57.81	99.53
**6c**	39.13	46.80	105.55
**6d**	72.78	65.65	98.59
**6e**	20.45	55.14	102.27
**6f**	54.58	63.23	102.19
**6g**	39.11	52.95	101.52
**6h**	50.37	66.45	93.73
**DTG**	96.89	82.61	99.57

**Table 3 molecules-29-01779-t003:** The half maximal inhibitory concentration (IC_50_, μM) of some compounds.

Compd No.	Cell Viability (100%), 48 h	
	Huh7	hepG2
**5c**	18.65 ± 0.53	18.28 ± 1.14
**5e**	2.64 ± 0.47	6.84 ± 0.99
**5f**	8.63 ± 0.33	12.06 ± 0.77
**5i**	12.26 ± 0.26	9.43 ± 0.70
**5k**	16.90 ± 0.48	15.18 ± 1.15
**5l**	7.62 ± 1.03	10.59 ± 0.67
**5p**	5.42 ± 0.43	4.83 ± 1.17
**5q**	7.85 ± 0.90	18.53 ± 2.85
**5t**	10.80 ± 0.85	11.16 ± 0.32
**5u**	12.63 ± 0.58	34.68 ± 5.13
**5v**	42.22 ± 1.10	24.75 ± 2.21
**5w**	15.91 ± 0.31	25.70 ± 0.42
**6c**	13.86 ± 0.81	27.75 ± 2.94
**6e**	16.22 ± 0.37	19.74 ± 2.19
**DTG**	>50	>50

## Data Availability

All relevant data are within the manuscript and Appendix A.

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
