# Peer review of "Discovery of Dolutegravir Derivative against Liver Cancer via Inducing Autophagy and DNA Damage"

_molecules, 2024, doi:10.3390/molecules29081779_

Round 1

Reviewer 1 Report

Comments and Suggestions for Authors

In this article, the authors introduced dolutegravir-1,2,3-triazole compounds through click chemistry and they claim that these compounds display remarkable inhibitory activities against two hepatocellular carcinoma cell lines such as Huh7 and HepG2. However, this manuscript lacks information in some aspects. The authors should provide detailed information on the selection of compounds in this study. Also, the article has a lot of redundancies that should be fixed wherever.   

specific comments

1. Page 2, line 59,  Include the full form of abbreviations when appear first time in the text 

2. The author must include a clear synthetic route on page 3 as a scheme 1 with detailed information such as reaction conditions, reagents, and time.

3. The relevant citations should be included in appropriate places.

4. In line 101, page 4 ---IC50--IC50

5. Some of the figures lacks details informaion ex; figure 3 and 8 and clear magnification of the image must be provided.

6. In SI file, the NMR needs improvement and should be rechecked again for example compound 5a ----13C NMR spectra values provided in copy and spectrums, as well clear experimental details should be provided with relevent citations. 

Comments on the Quality of English Language

minor checks typos need attention

Reviewer 2 Report

Comments and Suggestions for Authors

The author introduced terminal alkynes into the core structure of polytetravir and synthesized 34 novel polytetravir-1,2,3-triazole compounds through click chemistry. These compounds exhibit significant inhibitory activity on two liver cancer cell lines, Huh7 and HepG2. This job is very interesting and fulfilling. Therefore, I do not object to publishing this article on Molecules. However, I have some comments, as shown below. Revision is crucial for publication.

1. In Table 1, "n" can be deleted.

2. Methylene blue (MB) experiments should be conducted to study the production of ROS.

3. The IC50 differences between Huh7 and hepG2 should be discussed. What is the reason for this phenomenon?

4. Considering biocompatibility, compounds 5q and 6c have greater research value.

5. "Molecules 2023, 28 (2), 733" should be cited.

Reviewer 3 Report

Comments and Suggestions for Authors

The present manuscript by Xi-xi Hou et al describes the preparation of thirty four dolutegravir-1,2,3-triazole derivatives and their study as anticancer agents. Two of the compounds exhibited promising ability to inhibit the activity of liver cancer cells. Overall it is a well written paper with results which could be of interest to researchers in the field. In revising their manuscript the authors should take under consideration the following:

1) Compound 3 could be a mixture of two diastereomers. Where they separated and how the absolute configuration of 3 as depicted in Figure 2 was assigned? Please explain or give the appropriate reference.

2) An appropriate reference/control compound should be included/reported for all biological assays.

3) It was claimed that 5e exhibited low toxicity against normal cells. However neither an assay nor results are presented. These should be included in the manuscript.

4) Line 107, “Table 3. The half-maximal inhibitory concentration (IC50) of some compounds.” should change to “Table 3. The half-maximal inhibitory concentration (IC50, μM) of some compounds.”.

Round 2

Reviewer 1 Report

Comments and Suggestions for Authors

The author improved the manuscript quite better as well as similarity need to be improved by rephrasing. 

Comments on the Quality of English Language

english has to improve by rephrasing appropriate places and require slight modifications 

Author Response

Thank you, we have corrected the typo and format of our manuscript. Our manuscript will be thoroughly English-edited by the MDPI English-Editing team. 

Reviewer 3 Report

Comments and Suggestions for Authors

The authors satisfactorily responded to my three commends. Thus, I recommend publication of the manuscript without any farther revisions.

Author Response

Thank you very much for taking the time to review this manuscript.